# Hypoxia Alters the Expression of CC Chemokines and CC Chemokine Receptors in a Tumor–A Literature Review

**DOI:** 10.3390/ijms21165647

**Published:** 2020-08-06

**Authors:** Jan Korbecki, Klaudyna Kojder, Katarzyna Barczak, Donata Simińska, Izabela Gutowska, Dariusz Chlubek, Irena Baranowska-Bosiacka

**Affiliations:** 1Department of Biochemistry and Medical Chemistry, Pomeranian Medical University in Szczecin, Powstańców Wielkopolskich 72, 70-111 Szczecin, Poland; jan.korbecki@onet.eu (J.K.); d.siminska391@gmail.com (D.S.); dchlubek@pum.edu.pl (D.C.); 2Department of Anaesthesiology and Intensive Care, Pomeranian Medical University in Szczecin, Unii Lubelskiej 1, 71-281 Szczecin, Poland; klaudynakojder@gmail.com; 3Department of Conservative Dentistry and Endodontics, Pomeranian Medical University in Szczecin, Powstańców Wlkp. 72, 70-111 Szczecin, Poland; kasiabarczak@vp.pl; 4Department of Medical Chemistry, Pomeranian Medical University in Szczecin, Powstańców Wlkp. 72, 70-111 Szczecin, Poland; izagut@poczta.onet.pl

**Keywords:** hypoxia, hypoxia inducible factor, chemokine, cancer, NF-κB

## Abstract

Hypoxia, i.e., oxygen deficiency condition, is one of the most important factors promoting the growth of tumors. Since its effect on the chemokine system is crucial in understanding the changes in the recruitment of cells to a tumor niche, in this review we have gathered all the available data about the impact of hypoxia on β chemokines. In the introduction, we present the chronic (continuous, non-interrupted) and cycling (intermittent, transient) hypoxia together with the mechanisms of activation of hypoxia inducible factors (HIF-1 and HIF-2) and NF-κB. Then we describe the effect of hypoxia on the expression of chemokines with the CC motif: CCL1, CCL2, CCL3, CCL4, CCL5, CCL7, CCL8, CCL11, CCL13, CCL15, CCL16, CCL17, CCL18, CCL19, CCL20, CCL21, CCL22, CCL24, CCL25, CCL26, CCL27, CCL28 together with CC chemokine receptors: CCR1, CCR2, CCR3, CCR4, CCR5, CCR6, CCR7, CCR8, CCR9, and CCR10. To better understand the effect of hypoxia on neoplastic processes and changes in the expression of the described proteins, we summarize the available data in a table which shows the effect of individual chemokines on angiogenesis, lymphangiogenesis, and recruitment of eosinophils, myeloid-derived suppressor cells (MDSC), regulatory T cells (T_reg_), and tumor-associated macrophages (TAM) to a tumor niche.

## 1. Introduction

Chemokines are a group of almost 50 cytokines that constitutes one of the elements of the tumor microenvironment and intercellular communication. They activate over 20 receptors [1] and are divided into four families according to the shared N-terminal motif:- α chemokines with the CXC motif,- β chemokines with the CC motif,- γ chemokines with the XC motif,- δ chemokine with the CX3C motif.

The very name “chemokines,” an abbreviation for “chemotactic cytokines,” is associated with their ability to induce chemotaxis in various cells of the immune system and thus participate in the recruitment of these non-malignant cells to a tumor niche [2]. Chemokines are also crucial for the migration, invasion, and metastasis of cancer cells [3]. An important function of chemokines in cancer is to support the course of angiogenesis [4,5,6,7,8,9,10,11,12,13,14,15,16,17,18,19] and lymphangiogenesis [15,20,21,22,23,24]. Another important role of chemokines in cancer processes is the recruitment of cells to a cancer niche, in particular cancer-associated fibroblasts (CAF) [25,26], eosinophils [27,28,29], regulatory T cells (T_reg_) [30,31,32,33,34], T helper type 17 (Th17) [35,36,37], tumor-associated neutrophils (TAN) [38], tumor-associated macrophages (TAM) [11,39,40,41,42,43,44,45,46,47], and myeloid-derived suppressor cells (MDSC) [48]. However, some chemokines exert anti-cancer effects by causing tumor-infiltrating lymphocytes (TIL) infiltration, in particular CC motif chemokine ligand (CCL)2/monocyte chemoattractant protein (MCP)-1 [49,50,51,52,53,54], CCL7/MCP-3 [55], CCL3/macrophage inflammatory protein (MIP)-1α and CCL4/MIP-1β [56,57,58,59], CCL5/regulated on activation, normally T cell expressed and secreted (RANTES) [60,61,62,63,64,65,66], CCL17/thymus and activation regulated chemokine (TARC) and CCL22/macrophage-derived chemokine (MDC) [67,68], CCL19/EBI1-ligand chemokine (ELC) and CCL21/secondary lymphoid tissue chemokine (SLC) [69,70,71,72], CCL27/ESkine and CCL28/mucosae-associated epithelial chemokine (MEC) [73,74,75,76]. Finally, the expression of various chemokines is strictly regulated. Many of them are induced by inflammatory reactions and the transcription factor nuclear factor κB (NF-κB) [77,78]. No less important for changes in chemokine expression in a tumor are hypoxia inducible factors (HIF) [79] (Figure 1).

As there are no comprehensive reviews in the literature on the impact of hypoxia on the chemokine system in a tumor, the aim of this review is to collect all currently available and significant data on the impact of low oxygen levels on the expression of chemokines in the tumor. We focused on the β chemokines with the CC motif because of the variety of processes in which they are involved, including the multiple types of cells they recruit to the tumor niche. The work was divided into subsections discussing each receptor and its designated ligands, eschewing the traditional order in which chemokines are discussed one by one by their name. The logic behind our division is that each receptor is activated by many chemokines, which indicates their common properties.

## 2. Hypoxia: Intracellular Signaling

Hypoxia denotes an environment characterized by oxygen deficiency. In the early stages of tumor development, the blood vessels are unable to supply oxygen to inside a tumor [80]. This results in the formation of areas with chronic (continuous, non-interrupted) hypoxia. In these areas, cancer cells are selected for resistance to apoptosis and p53 protein dysfunction [81,82]. In further stages of tumor development, the tumor produces blood vessels with structural abnormalities that lead to periodic, cyclic oxygen deficiencies in various parts of the tumor [83]. This kind of hypoxia is known as cycling (intermittent, transient) hypoxia. Both types of hypoxia induce various changes in the expression of many genes that are important in the development of cancer as they lead to angiogenesis and migration of cancer cells [84]. The gene expression profile is similar between both types of hypoxia because of the activation of the same transcriptional factors in different ways.

### 2.1. Chronic Hypoxia

Chronic hypoxia alters the expression of more than 3500 genes [84,85]. HIF-1 alone directly changes the expression of more than 200 genes [85]. This is caused by the activation of HIF and their binding to the hypoxia response element (HRE) sequences within the promoters of respective genes. There are three HIF-HIF-1, HIF-2, and HIF-3, each consisting of two subunits. β subunits are not regulated by oxygen levels. In contrast, α subunits (HIF-1α, HIF-2α, and HIF-3α) are regulated by post-translational modification and increased expression at the transcriptional level. In normoxia, HIF-α is hydroxylated by prolyl hydroxylase domain enzymes (PHD) [86] and the factor inhibiting hypoxia-inducible factor (FIH) [87,88] (Figure 2). PHD hydroxylation of Pro^402^ HIF-1α, Pro^564^ HIF-1α, Pro^405^ HIF-2α, and Pro^531^ HIF-2α results in the ubiquitination by the von Hippel-Lindau protein (pVHL) [89,90,91,92,93], which leads to proteasomal degradation of HIF-α. At the same time, hydroxylation of Asn^803^ HIF-1α and Asn^847^ HIF-2α by FIH does not lead to degradation of HIF-α but prevents the interaction between HIF-α and CREB-binding protein (CBP) /p300 transcriptional coactivator [93,94].

PHD and FIH are enzymes that need oxygen for their activity [95]. For this reason, when the oxygen level decreases, the activity of these enzymes decreases. This, in turn, leads to a reduction in the hydroxylation of HIF-α thereby inhibiting proteasomal degradation of the protein. PHD are FIH enzymes that require different concentrations of oxygen [96]. For PHD, the Michaelis constant (K_m_) for oxygen is 230–250 μM [97] and 90 ± 20 μM for FIH [98]. The accumulation of HIF-α increases, which leads to an assembly of HIF complexes and an increase in the expression of genes dependent on these transcription factors. The maximum accumulation of HIF-1α occurs in the first 4 h of chronic hypoxia and then the level of this protein rapidly decreases [99,100]. In contrast, activation of HIF-2 and HIF-3 occurs after longer periods (2–3 days long) of uninterrupted chronic hypoxia [99,100]. Importantly, some types of tumors do not express HIF-2α in a cancer cell [101] and the increased expression level of HIF-2α may not occur in a cancer cell but in TAM [102].

In hypoxia, an increase in protein level of HIF-1α is associated not only with a reduction in the proteolytic degradation of the protein but also with increased expression at the transcriptional level, associated with the activation of NF-κB which binds to the *HIF1A* gene promoter [103,104,105]. The activation of NF-κB in chronic hypoxia occurs in many ways. In particular, under the hypoxia condition inhibitor of NF-κB kinase (IKK) activity increases by reducing the hydroxylation of IKKβ by PHD1 [106,107]. There is also an increase in Ca^2+^ in the cytoplasm, which results in the activation of calcium/calmodulin-dependent kinase 2 (CaMK2) and in consequence, the ubiquitination of IKKγ/NF-κB essential modulator (NEMO) [108]. This is followed by an increase in the activation of IKK, a kinase involved in the phosphorylation and degradation of inhibitor of NF-κB α subunit (IκBα). CaMK2 also indirectly causes IκBα sumoylation, which leads to the release and activation of NF-κB [108].

Chronic hypoxia also results in a change in gene expression that is not related to the activation of HIF and NF-κB but is related to changes in DNA and histone methylation [109]. A reduction in the amount of oxygen results in the decreased activity of the ten-eleven translocation protein (TET) and jumonji histone demethylases (JHDMs) [110,111], enzymes responsible for the demethylation of DNA and histones. Consequently, a hypoxia-induced increase in the methylation of DNA and histones leads to the reduced expression of many genes.

### 2.2. Cycling Hypoxia

Chronic hypoxia activates signaling pathways that cause angiogenesis [112]—a proliferation of blood vessels that supply the growing tumor with oxygen. Nevertheless, such vessels exhibit certain structural abnormalities [113] which results in their leakiness [114], and lack of a conventional hierarchy [114,115]. Because of this, blood in such vessels periodically changes its route [83], which results in tumor segmentation into hypoxia areas that re-oxygenate after some time [116] (Figure 3). This process loops every few minutes [116,117] for up to several hours [118,119]. This type of hypoxia is called cycling (intermittent, transient) hypoxia and is characteristic of a malignant tumor. The larger the tumor, the greater the fluctuations in oxygen concentration [120]. Also, the oxygen concentration fluctuation pattern is strictly characteristic of the cell line that has produced the tumor [116,118,120].

Both types of hypoxia, chronic and cycling, activate the same transcription factors and thereby change the expression of similar genes [84]. Nevertheless, they do differ significantly in the mechanism of activation of individual transcription factors. In cycling hypoxia, HIF-1α expression gets stronger and longer with subsequent cycles [121,122,123,124]. In chronic hypoxia the increase in HIF-α expression is dependent on the decrease in oxygen [86,92], whereas in cycling hypoxia it depends on the reactive oxygen species (ROS) [125]. Importantly, this type of hypoxia induces the degradation of HIF-2α and thereby reduces HIF-2 activity [126]. During reoxygenation, complete degradation of HIF-1α occurs [122]. Hypoxia induces the activation of HIF-1, which in turn increases the expression of PHD which remains inactive in hypoxic conditions. When reoxygenation occurs, PHD hydroxylate and degrade HIF-1α to a much greater degree than in normoxia [127,128,129].

Cycling hypoxia is also associated with the activation of NF-κB, in a ROS-dependent mechanism [125,130,131]. Importantly, cycling hypoxia induces a much higher expression of pro-inflammatory genes than chronic hypoxia [100,132,133,134]. This is one of the causes of chronic inflammation in a tumor that participates in the development of cancer [135].

## 3. The Effect of Hypoxia on the Chemokine System in a Tumor: β Chemokines with the CC Motif

β chemokines are a group of 28 chemotactic cytokines with a shared N-terminal CC motif [1]. These chemokines are ligands for ten classical receptors of β chemokines. They play a significant role in cancer processes (Table 1)—angiogenesis [4,5,6,7,8,9,10,11,12,13,14,15,16,17,18,19], lymphangiogenesis [15,20,21,22,23,24], and in the recruitment of tumor-associated cells [11,25,26,27,28,29,30,31,32,33,34,35,36,37,38,39,40,41,42,43,44,45,46,47,48]. Some of them are important in the infiltration of a tumor by TIL, which is why they have carcinogenic properties [49,50,51,52,53,54,55,56,57,58,59,60,61,62,63,64,65,66,67,68,69,70,71,72,73,74,75,76]. The nature of these properties depends on the tumor microenvironment.

This review describes the effects of hypoxia on the expression of only 22 representatives of the β chemokine family, as four of them (CCL6/C10, CCL9/CCL10/MIP-1γ, and CCL12/MCP-5) are murine chemokines not present in human tumors, along with a lack of data on the effect of hypoxia on the expression of CCL14/hemofiltrate CC chemokine (HCC)-1 and CCL23/CKβ8 in cancer and tumor-associated cells.

The influence of changes in the expression of a given chemokine on neoplastic processes should be analyzed on the basis of the information summarized in the Table 1. Also, an important consequence of an elevated expression of a particular chemokine is the induction of neoplastic cell migration [3]. The data presented below are mainly based on in vitro tests on different cell lines, which show the direct effect of hypoxia on the discussed neoplastic processes. However, it is a simplified model that significantly differs from the actual tumor microenvironment and hypoxic microenvironment. In a tumor, cells are affected not only by hypoxia, but also by other factors induced by hypoxia and acting together with hypoxia. An example of such a factor is cyclooxygenase-2 (COX-2) and the product of this enzyme activity, prostaglandin E_2_ (PGE_2_) [136,137,138]. Acidosis is also important, as it is largely caused by hypoxia and interactions with hypoxia [139,140,141]. 

### 3.1. Effect of Hypoxia on the Expression of CCR1 and the Ligands of This Receptor–CCL15/HCC-2 and CCL16/HCC-4

#### 3.1.1. Effect of Hypoxia on CCL15/HCC-2 Expression

Expression of CCL15/HCC-2 is not changed by chronic hypoxia as confirmed by the experiments on lung adenocarcinoma cells [13] and breast cancer cells [142], although chronic hypoxia may reduce CCL15/HCC-2 expression on the hepatocellular carcinoma cells [143] and primary human monocytes [144]. In contrast, TAM chronic hypoxia increases the expression of this chemokine, in a HIF-2-dependent manner [145]. 

Hypoxia/reoxygenation experiments showed an increase in CCL15/HCC-2 expression on human umbilical vein endothelial cells (HUVEC) [146]. This chemokine increased the CC motif chemokine receptor (CCR)1-dependent expression of intracellular adhesion molecule-1 (ICAM-1) on these endothelial cells resulting in increased monocyte adhesion. This mechanism may be important in cycling hypoxia-induced TAM recruitment, although further research is required to better understand the impact of cycling hypoxia on neoplastic processes involving CCL15/HCC-2 in hepatocellular carcinoma and colorectal cancer, i.e., organs with a high expression of CCL15/HCC-2 [147].

#### 3.1.2. Impact of Hypoxia on CCL16/HCC-4 Expression

CCL16/HCC-4 expression is increased under the influence of chronic hypoxia, as demonstrated on HepG2 cells [148], in a HIF-1-dependent mechanism. On the other hand, experiments on hepatocellular carcinoma of the MHCC-97L line [143] and lung adenocarcinoma cells [13] showed that chronic hypoxia does not change the expression of this chemokine. In general, the effect of hypoxia on neoplastic processes on hepatoma cancer via CCL16/HCC-4 is poorly researched.

#### 3.1.3. Impact of Hypoxia on CCR1 Expression

CCR1 is a receptor for many CC chemokines including CCL3/MIP-1α, CCL4/MIP-1β, CCL5/RANTES, CCL7/MCP-3, CCL11/Eotaxin-1, CCL13/MCP-4, CCL14/HCC-1, CCL15/HCC-2, CCL16/HCC-4, and CCL23/myeloid progenitor inhibitory factor (MPIF)-1 [1]. Its expression is regulated by chronic hypoxia via various mechanisms. Reduced oxygen levels decrease the activity of JHDMs [111], enzymes involved in histone demethylation, and thus hypoxia causes methylation of histones in the promoter of the *CCR1* gene [149]. This results in a reduced CCR1 expression in macrophages and monocytes [144,149]. The promoter of *CCR1* gene also contains HRE [150] and for this reason chronic hypoxia increases the expression of the CCR1 in THP-1 monocytes by HIF-1 [150] as well as in fresh human monocytes [151].

CCR1 expression can be increased in multiple myeloma plasma cells by HIF-2 [152]. This mechanism is crucial for the migration of these cells to CCL3/MIP-1α [152]. In contrast, a study on the breast cancer cells shows that chronic hypoxia does not affect CCR1 expression in cancer cells [153]. Therefore, it is crucial to conduct further research on the effect of hypoxia on the CCR1 expression in various tumor cells.

### 3.2. The Effect of Hypoxia on the Expression of the CCR2 Receptor and its Ligands: CCL2/MCP-1, CCL7/MCP-3, CCL8/MCP-2, and CCL13/MCP-4

#### 3.2.1. Impact of Hypoxia on CCL2/MCP-1 Expression

Many studies show that CCL2/MCP-1 expression is reduced or increased in response to reduced oxygen levels. Chronic hypoxia has been reported to decrease the expression of this chemokine in glioma cells [154,155], HUVEC [156], monocytes and macrophages [144,157,158], and in uveal melanoma [159]. In contrast, it has also been reported to increase in breast cancer cells [142], cervical squamous carcinoma cells [46], hepatoma cells [160], in another study on HUVEC [161], dermal fibroblasts [162], Lewis lung carcinoma cells [163,164], multiple myeloma [165], primary mouse astrocytes [166], and vascular endothelial cells [167,168]. These discrepancies are associated with the complex regulation of expression of this chemokine in hypoxia.

Chronic hypoxia is associated with a decrease in the activity of oxygen-dependent enzymes, for example, JHDMs [111]. The reduced activity of these enzymes leads to the methylation of the histones in the promoter and enhancer regions of the *CCL2* gene [149,169], especially in the NF-κB binding sequence [149]. This methylation leads to a decrease in CCL2/MCP-1 expression. At the same time, chronic hypoxia increases the activity of histone deacetylase 1 (HDAC1), which is associated with HDAC1 phosphorylation by protein kinase CK2 [170]. Then HDAC1 forms a complex with p65 NF-κB [171]. While NF-κB alone is a transcription factor that increases CCL2/MCP-1 expression, the HDAC1 acts as a transcriptional repressor [172]. The mechanism of silencing the CCL2/MCP-1 expression by this complex is associated with the deacetylation of histones.

The promoter of the *CCL2* gene contains HRE and so the expression of CCL2/MCP-1 is induced by HIF-1 [160,164,165,166,173,174], a significant accumulation of HIF-1α in chronic hypoxia results in elevated expression of CCL2/MCP-1. However, HIF-1α also binds to HDAC1, which offsets the repression of some genes caused by this protein [175]. It is likely that this is accompanied by increased CCL2/MCP-1 expression, but this is not supported by any research. Increased CCL2/MCP-1 expression in chronic hypoxia may also be related to NF-κB [162,164], although because of the interdependence of NF-κB and HIF-1 it is unknown whether NF-κB directly increases CCL2/MCP-1 expression or if it is via an increased HIF-1 expression [164]. Hypoxia may also indirectly affect the expression of CCL2/MCP-1. In MDSC, an increase in CCL2/MCP-1 expression in chronic hypoxia results from an increased expression of the regulator of G protein signaling-2 (Rgs2) [176].

The aforementioned pathways occur in chronic hypoxia. However, this type of hypoxia occurs only in a small section of the tumor [116]. The general level of CCL2/MCP-1 is affected by cycling hypoxia that also increases the expression of this chemokine, as shown by research on alveolar macrophages [177], THP-1 monocytes [178], endothelial cells [179], and melanoma cells [180]. This effect is dependent on the induction of NF-κB [162,177], in particular on the activation of this transcription factor by the cascades of mitogen-activated protein kinases (MAPK): extracellular signal-regulated kinase (ERK) and p38 [178,179,180]. It appears that in THP-1 monocytes, cycling hypoxia may increase CCL2/MCP-1 expression indirectly, via an increase in the expression of the receptor for advanced glycation end products (RAGE) [181,182,183], dependent on the activation of HIF-1 and NF-κB. CCL2/MCP-1 expression is then directly elevated by RAGE.

#### 3.2.2. Impact of Hypoxia on CCL7/MCP-3 Expression

Research on cervical squamous carcinoma cells [46], hepatocellular carcinoma [143], lung adenocarcinoma [13], and uveal melanoma [159] shows that chronic hypoxia does not affect CCL7/MCP-3 expression due to the lack of an HRE sequence in the promoter region of the *CCL7* gene [184]. On the other hand, a study on chondrocytes showed that overexpression of HIF-2α increased CCL7/MCP-3 expression [184], similar to acute myeloid leukemia cells that showed a hypoxia-induced increase release of CCL7/MCP-3 [185].

#### 3.2.3. Impact of Hypoxia on CCL8/MCP-2 Expression

The effect of chronic hypoxia and the expression of CCL8/MCP-2 depends on the experimental model. In hepatocellular carcinoma [143], lung adenocarcinoma [13], and uveal melanoma [159] hypoxia does not change the expression of CCL8/MCP-2. In contrast, in primary human monocytes, chronic hypoxia reduces the expression of CCL8/MCP-2 [144]. In addition, a study on acute myeloid leukemia cells showed that chronic hypoxia increases CCL8/MCP-2 release [185]. In another study on cervical squamous carcinoma cells, a chronic hypoxia-induced increase in CCL8/MCP-2 expression [46] is associated with an increase in the expression of zinc finger E-box-binding homeobox 1 (ZEB1), a transcription factor that increases the expression of this chemokine [46]. HIF-1 is directly responsible for increasing ZEB1 expression due to the presence of the HRE sequence in the *ZEB1* gene promoter [186,187]. This increase in CCL8/MCP-2 expression in cervical squamous carcinoma cells increases the recruitment of TAM to the hypoxic tumor niche [46].

#### 3.2.4. Impact of Hypoxia on CCL13/MCP-4 Expression

Chronic hypoxia does not change the expression of CCL13/MCP-4, as confirmed by research on breast cancer cells [142], hepatocellular carcinoma cells [143], and lung adenocarcinoma cells [13].

#### 3.2.5. Impact of Hypoxia on CCR2 Expression

CCR2 is the receptor of chemokines CCL2/MCP-1, CCL7/MCP-3, CCL8/MCP-2, and CCL13/MCP-4 [1]. Since the *CCR2* gene contains the HRE sequence [79], chronic hypoxia in macrophages increases CCR2 expression [79]. On the other hand, in M2 macrophages chronic hypoxia does not affect CCR2 expression [188] unless high-mobility group box 1 (HMGB1) is involved (nuclear chromatin-associated protein released by necrotic cells); then the expression of this receptor decreases in M2 macrophages [188].

In hypoxia, CCR2 expression is reduced in mouse L929 fibroblast cells [189] and human monocytes [79,144]. It has also been shown that chronic hypoxia disrupts the CCL2/MCP-1-dependent chemotaxis of monocytes [190], which is associated with a chronic hypoxia-induced increase in the expression of MAPK phosphatase 1 (MKP-1) [191] which reduces the activation of MAPK cascades and chemotaxis dependent on these signaling pathways [191].

Cycling hypoxia increases RAGE expression in monocytes [181,182,183], which increases CCR2 expression and enhances the effects of CCL2/MCP-1 on monocytes, in particular, increased migration and adhesion to vascular walls.

### 3.3. Effect of Hypoxia on the Production of CCR3 Receptor Ligands—CCL11/Eotaxin-1, CCL24/Eotaxin-2, and CCL26/Eotaxin-3

Chronic hypoxia increases CCL26/eotaxin-3 expression in ovarian cancer cells [192]. It also increases the expression of CCL26/eotaxin-3 and CCL24/eotaxin-2 in SK-Hep-1, Hep 3B HepG2, and MHCC-97L hepatocellular carcinoma [2,143], although in this tumor, this effect may depend on the model because, in another study, hypoxia in the HL-7702 and Huh-7 lines did not change the expression of these chemokines [193]. In cervical cancer cells, CCL11/eotaxin-1 expression is increased [28], although the effect of chronic hypoxia on the expression of this chemokine is only indirect. In breast cancer models, chronic hypoxia has been shown to increase CCL11/eotaxin-1 expression depending on the induction of oncostatin M expression [142]. In lung adenocarcinoma, chronic hypoxia cells do not increase the expression of any of the eotaxins [13]. All this shows that the effect of chronic hypoxia on the expression of eotaxins may depend on the tumor model.

Eotaxins participate in the recruitment of eosinophils to inflammation sites [194,195,196,197]. However, there is a lack of research showing that hypoxia, by increasing the expression of these chemokines, increases the recruitment of eosinophils to a tumor niche. This process appears to be secondary in hypoxia and depends on other chemokines, for example, CCL17/TARC [28] and so further research is required in this field. On the other hand, hypoxia has been shown to increase MDSC recruitment to hepatocellular carcinoma by increasing CCL26/eotaxin-3 expression [143]. Hypoxia also increases TAM recruitment to breast tumors by increasing CCL11/eotaxin-1 expression [142].

Chronic hypoxia does not change CCR3 expression on cancer cells. This was demonstrated by experiments on MDA-MB-231 and MDA-MB-435 breast cancer cells [153]. This indicates that chronic hypoxia does not increase the sensitivity of cancer cells to eotaxins.

### 3.4. Impact of Hypoxia on the CCL17/CCL22→CCR4 Axis

CCL17/TARC expression is affected by chronic hypoxia, although the nature of these changes depends on the adopted research model. In L929 mouse fibroblasts, chronic hypoxia reduces CCL17/TARC expression [189], while in human hepatoma cells [193] and lung adenocarcinoma cells [13], chronic hypoxia does not affect CCL17/TARC expression. In MDA-MB-231 breast cancer cells, hypoxia increases CCL17/TARC expression but not in the MCF-7 line [142]. In contrast, in HeLa and SiHa cervical cancer cells chronic hypoxia increases the expression of this chemokine [28], although in an indirect manner—via increased expression of thymic stromal lymphopoietin (TSLP), which in turn depends directly on HIF-1 [198]. Chronic hypoxia is also responsible for the recruitment of eosinophils (which participate in neoplastic processes) to a tumor niche dependent on CCL17/TARC expression increase.

CCL22/MDC expression is not changed by chronic hypoxia as shown by studies on hepatocellular carcinoma cells [143] and lung adenocarcinoma [13]. In contrast, cycling hypoxia has been shown to increase the expression of this chemokine in M2 macrophages [134]. However, more research is required to better understand the effect of cycling hypoxia on CCL22/MDC expression in cancer cells.

Chronic hypoxia also enhances the effect of the described chemokines on cervical and ovarian cancer cells by increasing the expression of the CCR4, CCL17/TARC, and CCL22/MDC receptor [192,199], the latter of which are responsible for an increase in the proliferation of cancer cells.

### 3.5. The Effect of Hypoxia on the Expression of the CCR5 Receptor and its Ligands: CCL5/RANTES CCL3/MIP-1α and CCL4/MIP-1β

#### 3.5.1. Effect of Hypoxia on the Expression of CCL5/RANTES

The *CCL5* gene contains HRE [79,153,184] and so CCL5/RANTES expression is increased under chronic hypoxia by HIF-1 [153,165] and HIF-2 [184]. It is also postulated that the expression of CCL5/RANTES under chronic hypoxia is elevated by NF-κB [200,201]. Nevertheless, some studies show that chronic hypoxia increases, and some that it reduces the expression of this chemokine. CCL5/RANTES expression is increased in acute myeloid leukemia cells [185], multiple myeloma cells [165], MDA-MB-231 and MDA-MB-435 breast cancer cells [142,153], cervical squamous carcinoma cells [46], and primary rat astrocytes [202].

The results of many studies indicate that chronic hypoxia does not affect or even reduce CCL5/RANTES expression. Although in some breast cancer lines (MDA-MB-231 and MDA-MB-435) chronic hypoxia increases CCL5/RANTES expression [142,153], it does not affect it in the MCF-7 line [142]. However, the experience of Voss et al. showed opposite results [203]. Chronic hypoxia increases CCL5/RANTES expression in MCF-7 cells but reduces it in MDA-MB-231, MDA-MB-435S, and MDA-MB-468 cells. Chronic hypoxia also does not affect CCL5/RANTES expression in hepatocellular carcinoma models [143,193], lung adenocarcinoma cells [13], and uveal melanoma cells [159]. The same lack of effect was demonstrated by studies on dermal fibroblasts [162]. Chronic hypoxia reduces CCL5/RANTES expression in activated NK cells, [204] and in monocytes [144]. In macrophages, results vary depending on the research model: chronic hypoxia may decrease [157,205], have no influence [158,177], or increase [201] CCL5/RANTES expression. Nevertheless, in cancer, CCL5/RANTES expression does occur, among others, in TAM [102,206,207,208], and for this reason, the mechanism of the effect of hypoxia on the expression of CCL5/RANTES in these cells requires further research.

In endothelial cells, CCL5/RANTES expression increases also in reoxygenation [146]. However, it has not yet been investigated whether this process depends on HIF or NF-κB or other signaling pathways.

The cited results show that the effect of hypoxia on CCL5/RANTES expression occurs through many mechanisms depending on the type of cell selected as the research model. However, the exact mechanisms of hypoxia-induced changes in CCL5/RANTES expression are not well understood.

#### 3.5.2. Effect of Hypoxia on the Expression of CCL3/MIP-1α and CCL4/MIP-1β

The effect of chronic hypoxia on CCL3/MIP-1α and CCL4/MIP-1β expression depends on the cell type. In non-cancer cells such as dermal fibroblasts [162] and primary hepatocytes [209] chronic hypoxia does not change the expression of CCL3/MIP-1α and CCL4/MIP-1β. However, in rat brain astrocytes, chronic hypoxia increases CCL3/MIP-1α expression [202]. In NK cells, chronic hypoxia reduces the expression of CCL3/MIP-1α and other pro-inflammatory cytokines [204], which contributes to cancer immunoevasion.

The effect of chronic hypoxia on CCL3/MIP-1α and CCL4/MIP-1β expression in a tumor cell depends on the type of tumor. In the cultured cervical squamous carcinoma cells [46], hepatocellular carcinoma [143,193] and lung adenocarcinoma [13], CCL3/MIP-1α expression is not elevated. However, in the breast tumor cells lines MDA-435S and MCF-7, chronic hypoxia increases the expression CCL3/MIP-1α [203]. Also in acute myeloid leukemia cells chronic hypoxia increases CCL3/MIP-1α release [185]. Chronic hypoxia has been shown to increase CCL4/MIP-1β expression in cervical cancer cells of the HeLa and SiHa line [28] and the release of this chemokine by acute myeloid leukemia cells [185]. In contrast, CCL4/MIP-1β expression is not increased by chronic hypoxia in cultured breast cancer cells [153], hepatocellular carcinoma [143,193] or, lung adenocarcinoma [13].

Chronic hypoxia in macrophages increases CCL3/MIP-1α expression [158,201,210] and CCL4/MIP-1β [205]. Nevertheless, this effect may depend on the duration of hypoxia, as 2-h chronic hypoxia reduces the expression of CCL3/MIP-1α and CCL4/MIP-1β macrophages [157]. The observed impact may also depend on the model. In THP-1 macrophages, chronic hypoxia does not affect CCL3/MIP-1α expression [205].

The effect of chronic hypoxia on the expression of both chemokines may be HIF-dependent. Since the *CCL3* gene promoter contains HRE [79], HIF-1 [165] and HIF-2 [184] increase the expression of CCL3/MIP-1α. The expression of this chemokine is also increased by NF-κB. The activation of this transcription factor during reoxygenation induces an increase in the CCL3/MIP-1α expression in cultured alveolar macrophages [177] and endothelial cells [146]. As a result, ICAM-1 expression increases on endothelial cells, which helps in the adhesion of monocytes and the recruitment of TAM to a tumor niche. It can be concluded from this that cycling hypoxia will increase CCL3/MIP-1α expression.

#### 3.5.3. Impact of Hypoxia on CCR5 Expression

Similar to the *CCL2* gene, chronic hypoxia results in an increase in histone methylation in the promoter of the *CCR5* gene [149]. It is associated with a decrease in the activity of JHDMs, enzymes that cause histone demethylation. The described effect of hypoxia causes a decrease in CCR5 expression in macrophages [149,210]. In human monocytes, chronic hypoxia does not affect CCR5 expression [151]. On the other hand, in NK cells chronic hypoxia reduces CCR5 expression at the mRNA level but does not change the level of the CCR5 protein [204]. Chronic hypoxia in monocytes and macrophages may also interfere with signal transduction from this receptor, which reduces CCL5/RANTES-dependent chemotaxis [190]. This is associated with a hypoxia-induced increase in the expression of MKP-1 [191], a phosphatase which reduces the activation of MAPK kinases.

The effect of chronic hypoxia on CCR5 expression depends on the model chosen. This is due to the complexity of regulating the expression of this protein. In cancer cells, such as breast cancer cells [153,211] and glioblastoma multiforme [205], chronic hypoxia increases CCR5 expression. This is due to the fact that the *CCR5* gene promoter contains HRE [153,211]. For this reason, CCR5 expression is increased by HIF-1 and HIF-2. This is important in cancer because the CCL5/RANTES→CCR5 axis is of great importance in the growth of tumors [212].

Chronic hypoxia reduces CCR5 expression in macrophages [149,210] and does not change expression in monocytes [151]. However, with cycling hypoxia, CCR5 expression increases and CCL5/RANTES effects on these cells, as shown by studies on monocytes of the THP-1 line [213].

### 3.6. Effect of Hypoxia on the Expression of CCL20 and Receptor CCR6

Chronic hypoxia increases CCL20/liver and activation-regulated chemokine (LARC) expression in a tumor. This chemokine increases indoleamine 2,3-dioxygenase (IDO) expression in TAM which leads to the accumulation of T_reg_ in a tumor niche [193]. However, the mechanism of the CCL20/LARC expression increase depends on the type of cancer. In glioma [214], hepatoma cancer cells [193] and ovarian cancer cells [192], chronic hypoxia increases CCL20/LARC expression. In a glioblastoma multiforme tumor, hypoxia-induced increased CCL20/LARC expression may also be found in astrocytes [215]. However, in MCF-7 and MDA-MB-231 breast cancer cells [142] and A549 and SPC-A1 lung adenocarcinoma [13] chronic hypoxia does not change the expression of this chemokine. Chronic hypoxia also increases CCL20/LARC expression in monocytes and macrophages [79,144,201,216]. In monocytes, this effect is greater due to HIF-1 dependent induction of CD300a expression [217]. 

The chronic hypoxia-induced CCL20/LARC depends on HIF and NF-κB. As the promoter of the *CCL20* gene contains HRE [79,193], this gene is induced under chronic hypoxia condition by HIF-1 [193]. Also at position 92/-82 in *CCL20* promoter there is an NF-κB-binding site [216,218] and so CCL20/LARC expression is increased by NF-κB under hypoxia condition. Experiments on primary human monocytes and macrophages show that in chronic hypoxia NF-κB, in particular p50 NF-κB homodimer, is crucial for increasing CCL20/LARC expression in these cells. 

CCL20/LARC expression is increased in response to cycling hypoxia in the SK-OV-3 ovarian adenocarcinoma cell line and PC-3 prostate cancer cell line [84].

The *CCR6* gene promoter contains HRE sequences [79]. However, there is a scarcity of published data on the induction of CCR6 expression by hypoxia. Available data show that by increasing CCL20/LARC expression, chronic hypoxia increases the activation of the CCL20/LARC→CCR6 axis [193].

### 3.7. Effect of Hypoxia on the CCL19/CCL21→CCR7 Axis

Chronic hypoxia increases CCR7 expression on breast cancer cells [219], epithelial ovarian carcinoma [220], head and neck cancers [221], and non-small cell lung cancer [222]. This is due to the fact that the *CCR7* gene promoter contains HRE and is thus induced by HIF-1 and HIF-2 [222]. At the same time, chronic hypoxia also increases the expression of CCR7 on macrophages [201] and NK cells [204] but not on immature DC [223]. The increase in CCR7 expression increases the migration of cancer cells [224,225]. Additionally, chronic hypoxia induces the expression of podoplanins in CAF [226] which increases the cancer cell migration in a CCR7-dependent manner.

An increase in CCR7 expression induces an increase in the expression of vascular endothelial growth factor (VEGF)-C and VEGF-D in tumor cells, i.e., growth factors that lead to lymphangiogenesis [15,20,22] (Figure 4). This facilitates the passage of cancer cells into the lymphatic vessels, which is associated with lymph node metastasis. It is estimated that from 1 g of tumor tissue, 1 million tumor cells are released daily into the bloodstream [227]. Metastasis is extremely inefficient and depends on many factors, particularly on the expression of chemokine receptors on cancer cells, including CCR7 associated with lymph node metastasis [219,228,229,230]. The lymph node has a high expression of CCR7 ligands: CCL19/ELC [231,232] and CCL21/SLC [233,234] and so these chemokines are responsible for the CCR7-dependent infiltration of lymphocytes to a lymph node [235]. The same mechanism causes the retention of cancer cells with CCR7 expression in a lymph node and the formation of metastasis to this peripheral lymphoid organ.

Chronic hypoxia causes an increase in mRNA expression of CCL19/ELC and CCL21/SLC in cervical cancer cells [46]. It appears that the effect of hypoxia on the expression of these chemokines is dependent on the type of cancer and whether it concerned mRNA or protein expression. Experiments on breast cancer cells show that chronic hypoxia only increases CCL21/SLC expression at the mRNA level, while at the protein level the expression of this chemokine is decreased [226]. On the other hand, in lung adenocarcinoma cells, chronic hypoxia does not change the expression of CCL19/ELC and CCL21/SLC [13]. Also in human hepatoma cancer chronic hypoxia cells do not increase CCL19/ELC expression [193]. In primary human monocytes, under the influence of chronic hypoxia, CCL19/ELC mRNA expression is reduced [144]. This indicates that hypoxia reduces CCL19/ELC and CCL21/SLC protein expression in a tumor but this effect is dependent on the tumor model.

### 3.8. Effect of Hypoxia on the Expression of CCL1/I-309 and CCL18/MIP-4, CCR8 Ligands

#### 3.8.1. Effect of Hypoxia on the Expression of CCL1/I-309

In a tumor cell, chronic hypoxia does not change the expression of CCL1/I-309 as demonstrated by lung adenocarcinoma cells [13] and hepatocellular carcinoma [143]. Nevertheless, in the L929 mouse fibroblast cell line, chronic hypoxia increases expression [189]. In the tumor, CCL1/I-309 expression occurs, among others, in CAF [236,237]. For this reason, further research is required on the effect of hypoxia on the expression of this chemokine in non-cancer cells such as CAF in order to better understand the role of the chemokine in cancer processes and in the hypoxic niche.

#### 3.8.2. Effect of Hypoxia on the Expression of CCL18/MIP-4

The *CCL18* gene promoter does not contain HRE [79] and so this gene is not induced by HIF [109,193]. Nevertheless, chronic hypoxia does induce a decrease in CCL18/macrophage inflammatory protein (MIP-4) expression in monocytes and macrophages [144], which is associated with increased histone methylation in the *CCL18* gene promoter [109]. This mechanism is dependent on a decrease in the activity of JHDMs, a histone demethylase that requires oxygen for its activity [111]. On the other hand, in lung adenocarcinoma cells [13] and hepatocellular carcinoma cells [143], hypoxia does not change the expression of CCL18/MIP-4, probably because of the low expression of this chemokine in cancer cells compared to TAM which have the highest synthesis of CCL18/MIP-4 in a tumor [102,238,239]. It seems that the impact of chronic hypoxia on the expression of CCL18/MIP-4 is only local. The expression of CCL18/MIP-4 is high in a tumor [240,241].

### 3.9. Effect of Hypoxia on the CCL25/TECK→CCR9 Axis in a Tumor

Chronic hypoxia does not affect CCL25/thymus-expressed chemokine (TECK) expression, as shown in lung adenocarcinoma cells [13] and hepatocellular carcinoma cells [143]. Nevertheless, the expression of CCL25/TECK occurs mainly in the gastrointestinal tract in physiological conditions [242]. Therefore research is needed on the effect of hypoxia on gastric cancer and colon cancer. Also, no studies are available on the effect of hypoxia on the expression of CCR9, the CCL25/TECK receptor.

### 3.10. Effect of Hypoxia Expression of the CCL28/MEC and CCL27/ESkin and their Receptor CCR10

#### 3.10.1. Effect of Hypoxia on the Expression of CCL27/ESkine

Chronic hypoxia does not affect CCL27/ESkine expression on the hepatocellular carcinoma cells model [143]. However, there are no other available studies of the effect of hypoxia on CCL27/ESkine, in particular on skin cells which show the highest expression of CCL27/ESkine [243].

#### 3.10.2. Effect of Hypoxia on the Expression of CCL28/MEC

Chronic hypoxia increases CCL28/MEC expression, as shown in cervical cancer cells [28], hepatocellular carcinoma [2,143], lung adenocarcinoma [13], melanoma cells [84], and ovarian cancer [192]. On the other hand, the experiment of Ye et al. showed that in hepatocellular carcinoma of the HepG2 line, hypoxia did not increase CCL28/MEC expression [193], which is contrary to the results of Ren et al. conducted on the same research model [2]. The effect of chronic hypoxia on CCL28/MEC expression has been shown to be HIF-1–dependent [192]. An increase in CCL28/MEC expression participates in the development of cancer, causing migration and invasion of hepatocellular carcinoma cancer cells [244]. At the same time, the effect of this chemokine may be increased by a hypoxia-induced increase in CCR10 expression in tumor cells [192]. An increase in CCL28/MEC expression in a hypoxic niche also leads to the recruitment of T_reg_ to this niche, in a process dependent on the CCR10 receptor on these cells [2,192]. T_reg_ participate in cancer immunoevasion and angiogenesis. CCL28/MEC also participates directly in angiogenesis, dependent on CCR3 on endothelial cells [13], which leads to the proliferation, migration, and tube formation of these cells. 

CCL28/MEC expression is increased in response to cycling hypoxia in the WM793B melanoma cell line [84].

#### 3.10.3. Effect of Hypoxia on the Expression of CCR10

Chronic hypoxia increases CCR10 expression, as demonstrated by studies on ovarian cancer cell models [192]. For this reason, it may increase the sensitivity of cancer cells to CCL28/MEC and CCL27/ESkine. 

## 4. Hypoxic Microenvironment as a Therapeutic Target

### 4.1. β Chemokines as a Therapeutic Target in Cancer Therapy

The quoted data show that, depending on the type of cancer, hypoxia may be neutral, or increase or reduce the expression of a given chemokine. In particular, in many types of cancer, hypoxia does not affect or decrease the expression of CCL2/MCP-1, CCL7/MCP-3, CCL5/RANTES, CCL3/MIP-1α, CCL4/MIP-1β CCL17/TARC, CCL22/MDC, CCL19/ELC, CCL21/SLC, CCL27/ESkine, and CCL28/MEC—chemokines that are relevant for the infiltration of tumors by cytotoxic and anticancer TIL [49,50,51,52,53,54,55,56,57,58,59,60,61,62,63,64,65,66,67,68,69,70,71,72,73,74,75,76]. For this reason, the increased expression of these chemokines through gene therapy will increase the anticancer response of the immune system and increase the effectiveness of immunotherapy, in particular for chemokines CCL2/MCP-1 [49,50,51,52,53,54], CCL7/MCP-3 [55], CCL3/MIP-1α and CCL4/MIP-1β [56,57,58,59], CCL5/RANTES [60,61,62,63,64,65,66,67,68,69,70,71,72,73,74,75,76,77,78,79,80,81,82,83,84], CCL17/TARC and CCL22/MDC [67,68], CCL19/ELC and CCL21/SLC [69,70,71,72], CCL27/ESkine and CCL28/MEC [73,74,75,76]. 

However, these chemokines have also pro-cancer properties and are subject to increased expression on some types of cancers. Therefore, on some models the inactivation of these chemokines does not have pro-cancer but anti-cancer effects. An example of this is CCL2/MCP-1, which causes apoptosis resistance and chemoresistance of cancer cells against anti-cancer drugs. For this reason the use of anti-cancer antibodies against CCL2/MCP-1 increases the effectiveness of anticancer therapy [245,246]. This chemokine also participates in the recruitment of TAM and MDSC (cells participating in tumor immune evasion and angiogenesis) and causes cancer cell migration. For this reason, inactivation of the CCL2/MCP-1→CCR2 axis by using anti-CCL2 antibodies or CCR2 antagonists increases the effectiveness of immunotherapy [247,248]. This therapeutic approach also shows anticancer properties in monotherapy [249,250,251,252,253,254]. Similarly, because of a significant pro-cancer effect, it is postulated that treatment may target the functioning of CCL3/MIP-1α [19,255], CCL5/RANTES [45,256]. Another therapeutic approach is the use of anti-CCR4 antibodies or CCR4 antagonists (receptor for CCL17/TARC and CCL22/MDC) in cancer therapy [257,258,259] to reduce the accumulation of T_reg_ and thus enhance the immunotherapy and anticancer response of the immune system.

Some researchers also postulate targeting the CCL20/LARC→CCR6 axis [260,261], which causes chemoresistance and migration of neoplastic cells, and so any disorder in the function of CCL20/LARC should increase the activity of anticancer drugs. The data presented above also show that hypoxia increases the expression of CCR7 and thus causes metastasis to lymph nodes. For this reason, one of the therapeutic approaches is targeting CCR7 to reduce metastasis to lymph nodes [262,263]. 

### 4.2. Cancer Therapy Targeting the Entire Hypoxic Microenvironment

Hypoxia causes changes in the expression of many β chemokines and their receptors. This leads to migration and invasion of cancer cells, which results in metastasis. An increase in the expression of the described chemokines leads to the angiogenesis, lymphangiogenesis, and recruitment of various cells to a tumor niche. The recruitment indirectly participates in all cancer processes, via the secretion of various factors by tumor-associated cells. Each of the aforementioned neoplastic processes (i.e., angiogenesis, lymphangiogenesis, and recruitment) may be caused by more than one chemokine and also factors other than chemokines. This entails considerable difficulties in designing new therapies because the cancer cells differ significantly within one tumor [264,265,266] and every tumor process is caused by many factors in different parts of the tumor. This is the result of genetic instability and selective pressure due to adverse conditions of the tumor microenvironment [81,82]. For this reason, therapy directed against only one chemokine may not give the desired results [267]. Another problem is the involvement of cancer processes in anti-cancer mechanisms. Many chemokines cause TIL infiltration, which has anti-cancer effects. However, the same chemokines recruit cells that promote the development of cancer, for example CCL17/TARC and CCL22/MDC [67,68,268,269,270,271,272], CCL20/LARC [37,273,274] or CCL19/ELC and CCL21/SLC [70,71,72,275,276]. Depending on the model, these chemokines cause either TIL infiltration or T_reg_ recruitment. The nature of the action of a given chemokine is more dependent on the balance of factors regulating the immune response in a tumor and not on the recruitment of individual cells to a tumor niche [277].

Given the aforementioned data collected from the literature on the subject, the best therapeutic approach in the treatment of cancer should not focus on a given single chemokine, but on the entire microenvironment that fosters neoplastic processes, including hypoxia. For this reason, new therapies are being developed to affect tumor cells that are in the hypoxic environment. In particular, prodrugs are being developed that are only activated in hypoxic conditions as well as substances decreasing the expression of HIF-1α or disturbing the regular function of HIF [278,279]. Then not only will HIF transcriptional activity be reduced but also the expression of many genes, including the discussed chemokines. The positive effects of anti-hypoxia treatment have also been shown in hyperbaric oxygen therapy which uses oxygen at a pressure higher than atmospheric pressure at the sea level [280,281]. Hyperbaric oxygen therapy has a strong preventive effect on hypoxia, common in many cancers [282,283,284,285]. It will affect all neoplastic mechanisms in the tumor, which in combination with other anti-cancer drugs may be an effective therapeutic approach against one of the most common human diseases [281,286]. 

The aforementioned therapies do not act on just one element of the hypoxic microenvironment but all factors induced by hypoxia, including the aforementioned chemokines. These therapies are not burdened with the disadvantages of monotherapies. In particular, they affect many mechanisms resulting in a single carcinogenic process. This makes this approach effective, even if two or more chemokines or other factors have the same effect.

## 5. Directions of Future Research

Almost all studies quoted in this review address the mechanisms inside the cancer cell. However, this model of research on cancer is very limited as currently a tumor is perceived as cancer cells interacting with non-cancer, albeit tumor-related, cells [287]. Therefore, in order to better understand all neoplastic processes discussed in this review, we need further research on the influence of chronic hypoxia on the expression of chemokines in cancer-related cells. In particular, this refers to TAM, T_reg_, MDSC, and other cells, and how these changes affect the cancer cell as well as other cancer-related cells. Finally, it needs to be emphasized that little is known about the influence of cycling hypoxia on chemokine expression.

## Figures and Tables

**Figure 1 ijms-21-05647-f001:**
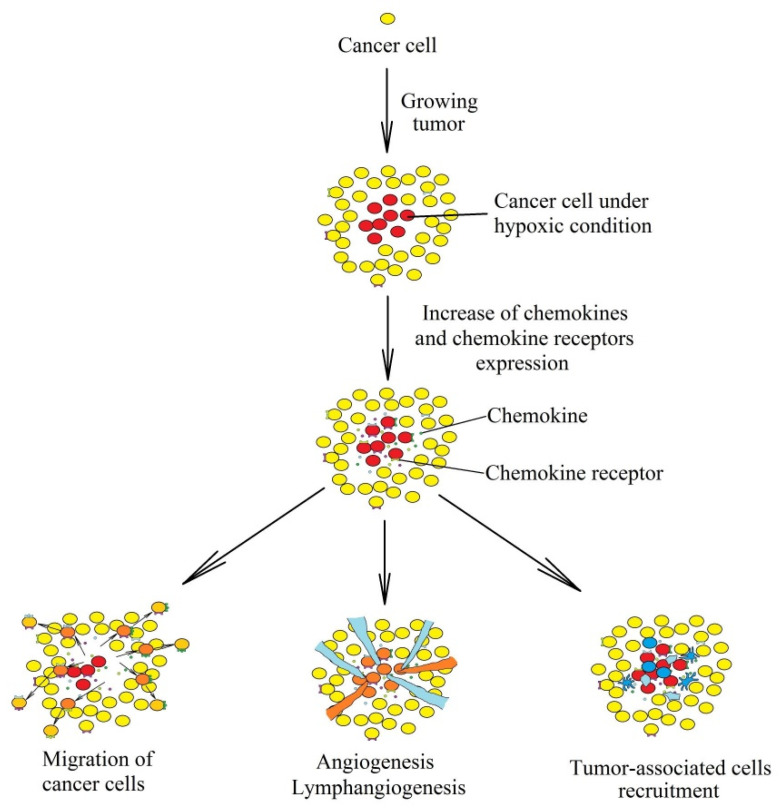
Consequences of the influence of hypoxia on the chemokine system in a tumor. A tumor or metastasis are initiated by a single cell that begins to divide uncontrollably. During the growth of the tumor, chronic hypoxia appears in the center of the tumor, which increases the concentration of chemokines in the tumor microenvironment and chemokine receptors on cancer cells. The activation of chemokine receptors on the cancer cell causes their migration. Chemokines cause angiogenesis, lymphangiogenesis, and recruitment of tumor-associated cells to areas with chronic hypoxia.

**Figure 2 ijms-21-05647-f002:**
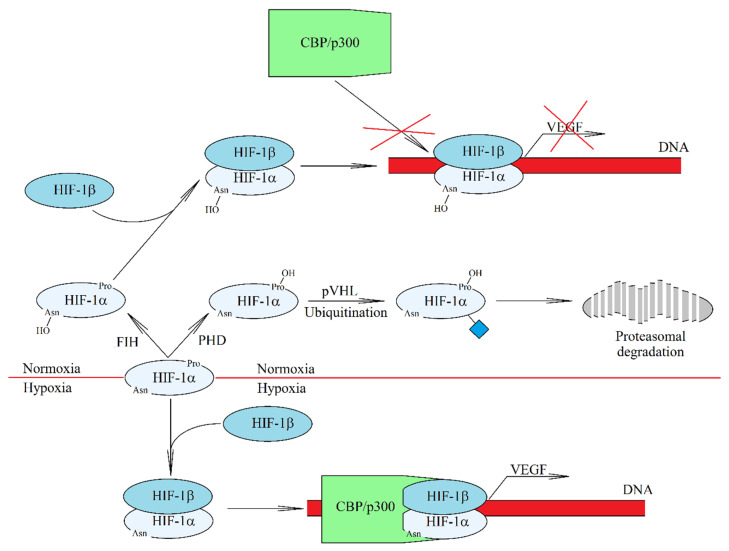
HIF-1 activation mechanism in hypoxia. In normoxia, HIF-1α is hydroxylated by oxygen-dependent enzymes. Hydroxylation by hypoxia-inducible factor (FIH) prevents the interaction between HIF-1α and CBP/p300 transcriptional coactivator. In turn, hydroxylation by prolyl hydroxylase domain (PHD) leads to the ubiquitination of HIF-1α and then proteasomal degradation of this HIF-1 subunit. As a result no HIF-1 is formed. Hypoxia, on the other hand, induces an increase in the level of HIF-1α, which is no longer hydroxylated by either FIH or PHD. A functional HIF-1 complex is formed and transferred to the cell nucleus from where it is responsible for the transcription of genes induced by hypoxia.

**Figure 3 ijms-21-05647-f003:**
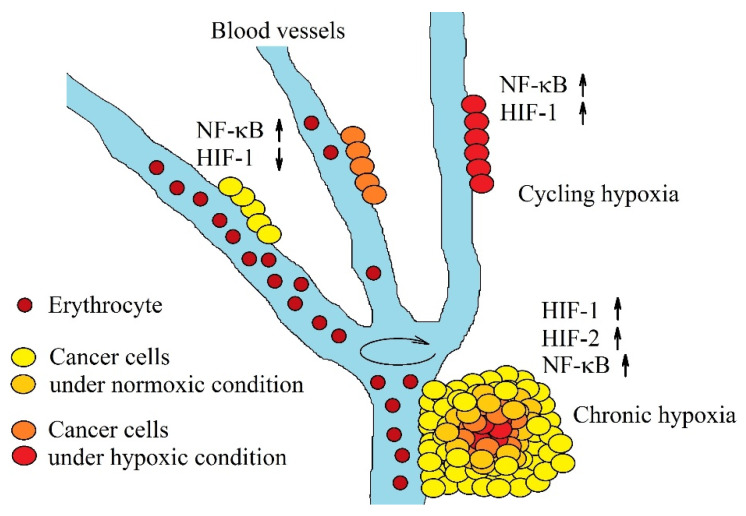
Two types of hypoxia in a tumor. Chronic hypoxia is associated with a long distance between the cells and the blood vessels that supply them with oxygen. This hypoxia induces an increase in activation of HIF-1 and HIF-2 and expression of hypoxia-inducible genes. The second type of hypoxia, cycling hypoxia, consists in blood flow through a given blood vessel in repeated cycles. The hypoxic stage is accompanied by the activation of NF-κB, and the activation of HIF-1. Reoxygenation is associated with the proteolytic degradation of HIF-1α and decrease in HIF-1 activation.

**Figure 4 ijms-21-05647-f004:**
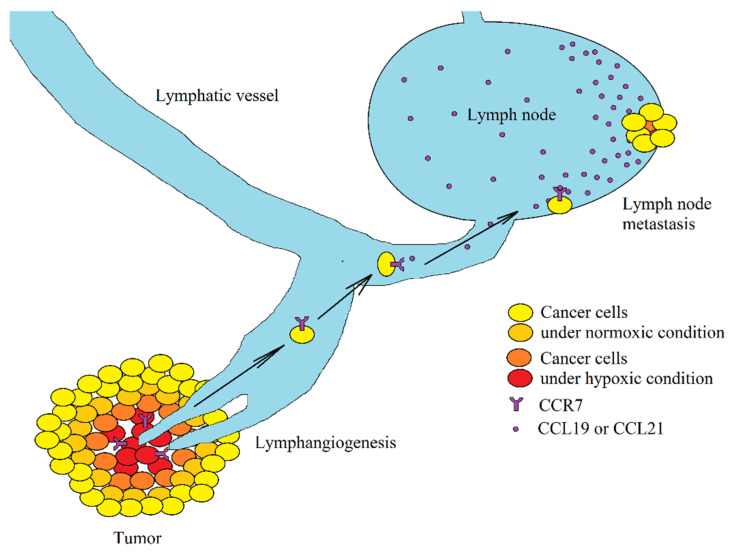
Association between hypoxia and lymph node metastasis. Hypoxia induces an increase in CCR7 expression on cancer cells. If such a cell enters lymph vessels, it will be retained in the lymph node because of a high concentration of ligands of this receptor (CCL19/ELC and CCL21/SLC) in the lymph node. This results in a lymph node metastasis.

**Table 1 ijms-21-05647-t001:** Characteristics of the chemokines discussed in this review, including the impact on the recruitment of cells to the tumor niche.

Chemokine	Alternative Name of the Chemokine	Receptor	Type of the Cell Recruited to the Tumor Niche	Effect on the Vascularization of the Tumor
CCL1	I-309	CCR8	TAM, T_reg_	Angiogenesis
CCL2	MCP-1	CCR2, CCR4	MDSC, TAM, Th17, TIL, T_reg_	Angiogenesis
CCL3	MIP-1α	CCR1, CCR5	CAF, MDSC, TIL, T_reg_	An increase in VEGF expression leading to angiogenesis
CCL4	MIP-1β	CCR1, CCR3, CCR5	CAF, MDSC, TIL, T_reg_	Increase in VEGF and VEGF-C expression leading to angiogenesis and lymphangiogenesis
CCL5	RANTES	CCR1, CCR3, CCR5	MDSC, TAM, TIL, T_reg_	An increase in VEGF expression leading to angiogenesis
CCL7	MCP-3	CCR1, CCR2, CCR3, CCR5	TAM, TIL	
CCL8	MCP-2	CCR3, CCR5	TAM, T_reg_	
CCL11	Eotaxin-1	CCR3, CCR5	Eosinophile	Angiogenesis
CCL13	MCP-4	CCR1, CCR2, CCR3		
CCL15	HCC-2, MIP-1δ, MIP-5, leukotactin-1	CCR1, CCR3	MDSC, TAM, TAN	Angiogenesis
CCL16	HCC-4, LEC, LCC-1, MTN1	CCR1, CCR5, CCR8		Angiogenesis
CCL17	TARC	CCR4	Eosinophile, Th17, TIL, T_reg_	
CCL18	PARC, MIP-4	PITPNM3, CCR8	T_reg_	Angiogenesis
CCL19	MIP-3β, ELC	CCR7	TIL, T_reg_	Increase in expression of VEGF-A, VEGF-C, and VEGF-D leading to angiogenesis and lymphangiogenesis
CCL20	MIP-3α, LARC, exodus-1	CCR6	Th17, T_reg_	Angiogenesis
CCL21	SLC	CCR7	TIL, T_reg_	Increase in expression of VEGF-A, VEGF-C, and VEGF-D leading to angiogenesis and lymphangiogenesis
CCL22	MDC	CCR4	Eosinophile, Th17, TIL, T_reg_	
CCL24	Eotaxin-2	CCR3	Eosinophile	Angiogenesis
CCL25	TECK	CCR9		
CCL26	Eotaxin-3	CCR3, CX3CR1	Eosinophile, MDSC, TAM	Angiogenesis
CCL27	ESkine	CCR10	TIL,	Lymphangiogenesis
CCL28	MEC	CCR3, CCR10	TIL, T_reg_,	Angiogenesis, lymphangiogenesis

CAF—cancer-associated fibroblasts; MDSC—myeloid-derived suppressor cells; TAM—tumor-associated macrophages; TAN—tumor-associated neutrophils; Th17—T helper type 17; TIL—tumor-infiltrating lymphocytes; T_reg_—regulatory T cells.

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
