# Peer review of "Hypoxia Alters the Expression of CC Chemokines and CC Chemokine Receptors in a Tumor–A Literature Review"

_ijms, 2020, doi:10.3390/ijms21165647_

Round 1

Reviewer 1 Report

This is a good manuscript and the authors well summarized the effect of tumor microenvironment hypoxia on chemokines, in particular b chemokines.  It’s novel that focusing on hypoxia and chemokine system in the tumor as. It’s also a lot of work to collect and summarize all of the data available about hypoxia and b chemokines.

There are some minor comments.

  1. Chemokines and their cognate receptors connect cancer and stromal cells and the tumor microenvironment. Recent studies indicate that inhibiting the chemokine system in cancer as a monotherapy or in combination with canonical or immune-mediated therapies could become future targets for immunotherapy. In my view, it would be very interesting if the authors could talk about this, and strength the impact of this manuscript and attract more readers.
  2. Some inaccurate sentences should be revisited. For examples, in line #93, the authors say chronic hypoxia regulates 8635 genes and this is mostly caused by the activation of HIFs and their binding to HRE at the promoters, this is not accurate, first, these eight thousand genes are only found in three cell lines, second, the reference paper never states that most of these genes harbor HREs and their gene expression is directly regulated by HIFs. Actually, people mostly agree that HIF regulates several hundred genes instead of eight thousand genes. In line #101, FIH does not simply interfere the interaction between HIF-α and HIF1b. FIH catalyzes asparagine hydroxylation, which blocks association of HIF-α transcription factors with CBP/p300 transcriptional coactivators. The authors also should revisit Figure 2. It’s also necessary to point out the different types of hydroxylase by PHD or FIH in Figure 2, current version look the same.
  3. It would be good to have a short introduction after section ‘3.’ and before section ‘3.1’.
  4. It’s better to write ‘in this review’ instead of ‘in this paper’.
  5. In line # 409, MDAMB 231 should be MDA-MB-231.

Author Response

Rev.1.

This is a good manuscript and the authors well summarized the effect of tumor microenvironment hypoxia on chemokines, in particular b chemokines.  It’s novel that focusing on hypoxia and chemokine system in the tumor as. It’s also a lot of work to collect and summarize all of the data available about hypoxia and b chemokines.

There are some minor comments.

1. Chemokines and their cognate receptors connect cancer and stromal cells and the tumor microenvironment. Recent studies indicate that inhibiting the chemokine system in cancer as a monotherapy or in combination with canonical or immune-mediated therapies could become future targets for immunotherapy. In my view, it would be very interesting if the authors could talk about this, and strength the impact of this manuscript and attract more readers.

Answer:An appropriate fragment has been added.

2. Some inaccurate sentences should be revisited. For examples, in line #93, the authors say chronic hypoxia regulates 8635 genes and this is mostly caused by the activation of HIFs and their binding to HRE at the promoters, this is not accurate, first, these eight thousand genes are only found in three cell lines, second, the reference paper never states that most of these genes harbor HREs and their gene expression is directly regulated by HIFs. Actually, people mostly agree that HIF regulates several hundred genes instead of eight thousand genes. In line #101, FIH does not simply interfere the interaction between HIF-α and HIF1b. FIH catalyzes asparagine hydroxylation, which blocks association of HIF-α transcription factors with CBP/p300 transcriptional coactivators. The authors also should revisit Figure 2. It’s also necessary to point out the different types of hydroxylase by PHD or FIH in Figure 2, current version look the same.

Answer:Corrected according to the reviewer's comments.

3. It would be good to have a short introduction after section ‘3.’ and before section ‘3.1’.

Answer:We have added a short fragment describing the CC chemokine. We have moved the table on the significance of CC chemokine to section 3.

4. It’s better to write ‘in this review’ instead of ‘in this paper’.

Answer:Corrected according to the reviewer's comment.

5. In line # 409, MDAMB 231 should be MDA-MB-231.

Answer:Corrected according to the reviewer's comment.

The description of the Fig. 1 has been extended.

Reviewer 2 Report

This manuscript is written in a very clear English language and the figures are well addressing the described processes. The manuscript is structured quite well. Also, the review is done very thoroughly.

In my opinion, this manuscript provides limited scientific value in general. As authors state themselves, the review describes collected data on the impact of hypoxia on the expression of chemokines in the tumor. Also, this manuscript explains the chronic and cyclic hypoxia. However, it lacks the more detailed discussion about what is the hypoxia itself, which levels of oxygen are considered to be hypoxic. Hypoxia in a tumor is very often closely associated with a more acidic environment (acidosis), these processes usually are going together. Thus it should be worthwhile to discuss if acidosis somehow affects the expression of chemokines.

The discussion seems quite superficial to me. Is the title of that section correct? As I understood from the discussion, authors are talking about hyperoxia (not hypoxia itself) as a therapy. Usually, not hypoxia itself is a target, but specific proteins under hypoxia are exploited as targets for chemical compounds, nanoparticles, antibodies, etc. In my opinion, it should be more focused on chemokines and their receptors as the main players in cancer treatment, with less focus on oxygen therapy, or at least with a more detailed explanation of how such treatment is related to chemokines.

Also, the quality of the figures could be enhanced. Not all image parts are explained in Figure 1.

Author Response

Rev.2.

This manuscript is written in a very clear English language and the figures are well addressing the described processes. The manuscript is structured quite well. Also, the review is done very thoroughly.

1. In my opinion, this manuscript provides limited scientific value in general. As authors state themselves, the review describes collected data on the impact of hypoxia on the expression of chemokines in the tumor. Also, this manuscript explains the chronic and cyclic hypoxia. However, it lacks the more detailed discussion about what is the hypoxia itself, which levels of oxygen are considered to be hypoxic.

Answer:We have added the levels of oxygen which result in decreased FIH and PHD activity.

2. Hypoxia in a tumor is very often closely associated with a more acidic environment (acidosis), these processes usually are going together. Thus it should be worthwhile to discuss if acidosis somehow affects the expression of chemokines.

Answer:We agree with the reviewer that acidosis is also an important element of the cancer microenvironment. Hypoxia causes acidosis because the expression of some glycolysis enzymes is increased by HIF-1. For this reason hypoxia enhances the Warburg effect, which causes the co-occurrence of hypoxia and acidosis in the same tumor areas. However, when studying the influence of hypoxia on the expression of proteins, scientists always use an in vitro model. Cancer cells are cultured under hypoxic condition in a medium that contains a buffer. For this reason, researchers only study the effect of hypoxia on the expression of chemokines, with a negligible effect of acidosis. Our review is based only on those works, and so we focused on the effect of hypoxia on the expression of chemokines without the influence of acidosis.

      Although we consider the influence of acidosis to be noteworthy, when we searched PubMed with a phrase: “chemokine and (acidosis or lactic or (Warburg effect)) and (cancer or tumor)”, we found only a few papers on CC chemokines: 2 papers on monocytes/macrophages, 1 on normal epithelial cells, 1 on normal fibroblasts, and only 1 on cancer cells (osteosarcoma cells), by Avnet S, Di Pompo G, Chano T, Errani C, Ibrahim-Hashim A, Gillies RJ, Donati DM, Baldini N. (Cancer-associated mesenchymal stroma fosters the stemness of osteosarcoma cells in response to intratumoral acidosis via NF-κB activation. Int J Cancer. 2017 Mar 15;140(6):1331-1345. doi: 10.1002/ijc.30540.). The latter study investigated 3 lines of that cancer, which reacted differently to acidosis. Some showed increased expression of CCL2 and CCL5, and other showed decreased expression.

      There are more papers on the effect of acidosis on the expression of CXC chemokines, but we did not deal with this family of chemokines in our paper. Therefore, it would be difficult to add information on the effect of acidosis on the expression of CC chemokines based on such a small number of papers, most of which deal with non-cancer cells. Any information based on such a scarce body of research would be too speculative.

      However, in the section 3 (before the section 3.1) we now describe the complexity of the tumor microenvironment and the fact that hypoxia increases various cancerogenic factors, which act together with hypoxia, especially acidosis. That is why the cited article should be analyzed with great caution, because it shows a simplified model.

3. The discussion seems quite superficial to me. Is the title of that section correct? As I understood from the discussion, authors are talking about hyperoxia (not hypoxia itself) as a therapy. Usually, not hypoxia itself is a target, but specific proteins under hypoxia are exploited as targets for chemical compounds, nanoparticles, antibodies, etc. In my opinion, it should be more focused on chemokines and their receptors as the main players in cancer treatment, with less focus on oxygen therapy, or at least with a more detailed explanation of how such treatment is related to chemokines.

Answer:The title of the subsection on therapy has been changed and divided into two part. In the first part, according to the reviewer's comments, we have discussed the use of chemokine and substances disrupting the functioning of chemokines in cancer therapy. The fragment about hyperoxia has been shortened. However, many papers suggest that a simultaneous increase and decrease in the expression of different chemokines (CCL2, CCL5) may have a anti-cancer effect. In short, chemokines recruit cells to a tumor niche. What happens to those cells, for example, whether monocytes will differentiate into M2 or M1, depends on the tumor microenvironment. This especially concerns the in vivo models in patients, where there is a functional immune system, contrarty to experimental animals. Therefore, in the second subsection on therapy, we suggest that maybe it would be better to target the entire hypoxic microenvironment.

4. Also, the quality of the figures could be enhanced. Not all image parts are explained in Figure 1.

Answer:The description of the Fig. 1 has been extended.